# Concurrent Topology Optimization for Maximizing the Modal Loss Factor of Plates with Constrained Layer Damping Treatment

**DOI:** 10.3390/ma15103512

**Published:** 2022-05-13

**Authors:** Zhanpeng Fang, Lei Yao, Junjian Hou, Yanqiu Xiao

**Affiliations:** Henan Engineering Research Center of New Energy Vehicle Lightweight Design and Manufacturing, Zhengzhou University of Light Industry, Zhengzhou 450002, China; 2016008@zzuli.edu.cn (L.Y.); 2014042@zzuli.edu.cn (J.H.); xiaoyanqiu@zzuli.edu.cn (Y.X.)

**Keywords:** topology optimization, concurrent design, constrained layer damping, modal loss factor, sensitivity analysis, representative volume element

## Abstract

Damping performance of the plates with constrained layer damping (CLD) treatment mainly depends on the layout of CLD material and the material physical properties of the viscoelastic damping layer. This paper develops a concurrent topology optimization methodology for maximizing the modal loss factor (MLF) of plates with CLD treatment. At the macro scale, the damping layer is composed of 3D periodic unit cells (PUC) of cellular viscoelastic damping materials. At the micro scale, due to the deformation of viscoelastic damping material affected by the base and constrained layers, the representative volume element (RVE) considering a rigid skin effect is used to improve the accuracy of the effective constitutive matrix of the viscoelastic damping material. Maximizing the MLFs of CLD plates is employed as the design objectives in optimization procedure. The sensitivities with respect to macrodesign variables are formulated using the adjoint vector method while considering the contribution of eigenvectors, while the influence of macroeigenvectors is ignored to improve the computational efficiency in the mesosensitivity analysis. The macro and meso scales design variables are simultaneously updated using the Method of Moving Asymptotes (MMA) to find concurrently optimal configurations of constrained and viscoelastic damping layers at the macro scale and viscoelastic damping materials at the micro scale. Two rectangular plates with different boundary conditions are presented to validate the optimization procedure and demonstrate the effectiveness of the proposed concurrent topology optimization approach. The effects of optimization objectives and volume fractions on the design results are investigated. The results indicate that the optimized layouts of the macrostructure are dependent on the objective mode and the volume fraction on the meso scale. The optimized designs on the meso scale are mainly related to the objective mode. By varying the volume fraction on the macro scale, the optimized designs on the meso scale are different only in their detailed size, which is reflected in the values of the equivalent constitutive matrices.

## 1. Introduction

Viscoelastic damping materials are often used to reduce the vibration and noise radiation of plate and shell structures. In particular, constrained layer damping (CLD) treatment has the advantages of simple implementation, low cost and high damping capability, and it has been widely used in the automobile, aviation, aerospace and naval industries [1]. To design lightweight structures with high damping performance, it is desirable to optimize the layout of the viscoelastic damping material in order to improve damping efficiency.

The topology optimization method was originally developed to find the optimized structural layout under given constraints [2]. The modal loss factor (MLF) is always used to evaluate the damping characteristics of the structure with viscoelastic damping treatment, and it can be defined as an objective function to optimize the layout of the viscoelastic damping material. Zheng et al. [3] utilized the Method of Moving Asymptotes (MMA) for maximizing the MLF to optimize the distribution of the viscoelastic damping material in a plate with CLD treatment. Kim et al. [4] compared the optimization results obtained via topology optimization to the strain energy distribution method and the mode shape method and pointed out that topology optimization is the most effective way to design the optimal damping layout in a viscoelastic damping structure. Yamamoto et al. [5] optimized the layout of damping material to maximize MLFs, which is expressed approximately by using the corresponding real eigenvalue. Madeira et al. [6,7] presented a multiobjective optimization approach to find the distribution of CLD material to minimize weight and maximize MLF simultaneously. Delgado and Hamdaoui [8] used the level set method (LSM) to perform the topology optimization of frequency-dependent viscoelastic structures to maximize MLF. Zhang et al. [9] proposed an improved Evolutionary Structural Optimization (ESO) to optimize the layout of CLD material for the vibration suppression of an aircraft panel. Zhang et al. [10] presented a two-level optimization method to design position layouts and thickness configurations of CLD materials to reduce the sound power of vibrating structures. 

Meanwhile, optimizing the distribution of viscoelastic damping materials to minimize vibration response and sound radiation has received attention from many scholars. Zhang and Kang [11] optimized the layout of damping layers in plate and shell structures to minimize sound radiation under harmonic excitations. Zheng et al. [12] presented the topology optimization of CLD treatment attached to thin plates to reduce sound radiation at low frequency resonance, and the effectiveness of the method was verified through numerical examples and experiments. Based on complex dynamic compliance, Takezawa et al. [13] developed an optimization methodology for damping material distribution to reduce the resonance peak response. Ma and Cheng [14] proposed a general methodology to find the optimal layout of viscoelastic damping layer for reducing the sound radiation of an acoustic black hole structure through topological optimization. Zhang and Chen [15] investigated the topology optimization of a damping layer under harmonic excitations and discussed the influences of the excitation frequency and the damping coefficients of the damping material on the distribution of the damping layer. 

Since the physical properties of the viscoelastic damping layer have a great influence on the damping performance, there is a great desire to optimize the microstructures of the damping layer with desirable properties [16]. Sigmund [17,18] first presented the inverse homogenization method to design materials with prescribed constitutive parameters. Huang et al. [19] used the bi-directional evolutionary structural optimization (BESO) method to design microstructures of two-phase material, which is composed of elastic material with high stiffness and viscoelastic material with high damping. Chen and Liu [20] proposed a multi-scale optimization method for the design of the microstructures of a viscoelastic damping layer to maximize MLFs. Asadpoure et al. [21] proposed a topology optimization framework to design multiphase cellular materials for improving damping characteristics under wave propagation. Yun and Youn [22] studied the optimal microstructure of viscoelastic damping material in sandwich structures subject to impact loads by using a microstructural topology optimization method. Liu et al. [23] utilized the BESO method to optimize the microstructure of viscoelastic materials with the aim of improving the MLF and frequency of macrostructures. Giraldo-Londoo and Paulino [24] presented a microstructural topology optimization approach to design the microstructure of multiphase viscoelastic composites to enhance energy dissipation characteristics. Zhang et al. [25] proposed a topology optimization method to find the optimal two-phase damping material layout in micro scales to make the composite materials with high stiffness and high broadband damping.

However, the above works concerning the topology optimization of viscoelastic damping material are concentrated on a one-scale design problem. With the development of optimization algorithms dealing with large-scale optimization problems [26], the idea of concurrent design was introduced into topology optimization while considering both the macro and micro scale to pursue a higher structural performance. Niu et al. [27] and Zuo et al. [28] presented a multi-scale design approach to maximize the natural frequency of the structure. Coelho et al. [29] presented a hierarchical structural optimization method for the simultaneous optimization of the structure and material of bi-material composite laminates, in order to minimize the structural compliance. Based on the ordered Solid Isotropic Material with Penalization (SIMP) interpolation, Zhang et al. [30] proposed a multiscale topology optimization method to simultaneously optimize the macrostructural topology and configurations of microstructures. Gao et al. [31] developed dynamic multiscale topology optimization for the concurrent design of composite macrostructures and multiple microstructures to improve structural performance. Hoang [32] developed a multiscale topology optimization approach for lattice structures using adaptive geometric components, which consist of macromoving bars and the microbar. Zhang et al. [33] proposed a multiscale topology optimization method to minimize the frequency response of a two-scale cellular composite with spatially varying connectable graded microstructures. 

At present, the concurrent topology optimization of viscoelastic damping structures is still limited. Zhang et al. [34] presented a concurrent topology optimization method for the optimal layout on both macro and micro scales of the free-layer damping structures with damping composite materials. The damping layer is composed of 2D periodic damping material, which consists of a stiff damping material and a soft damping. The effective complex constitutive matrix of the damping composite materials are obtained using the classical homogenization method. In the above works, the viscoelastic material was seen as ‘free’ material when the equivalent constitutive matrix was calculated using the homogenization method. However, the viscoelastic damping layer in the CLD structure is constrained by the base and the constrained layers. The deformation of the viscoelastic damping layer is affected by the skins, which will lead to larger out-of-plane shear moduli than those obtained by neglecting the skin effect [35]. Hence, it is necessity to consider the skin effect when the effective material properties of the viscoelastic damping layer are estimated. 

The purpose of this work is to develop a concurrent topology optimization method for maximizing MLF of plates with CLD treatment. The plates with CLD treatment dissipate vibration energy through transverse shear strains induced in the viscoelastic damping layer, and the effective transverse shear moduli are the main focus. Therefore, it is assumed that the macrostructure of the damping layer is composed of the 3D periodic unit cells (PUC). The representative volume element (RVE) considering a rigid skin effect is used to improve the accuracy of the effective constitutive matrix of the viscoelastic damping material. A mathematical optimization model is established while maximizing MLF as the design objective. The sensitivities with respect to macrodesign variables are formulated using the adjoint vector method while considering the contribution of eigenvectors, while the influence of macroeigenvectors is ignored to improve the computational efficiency in the mesosensitivity analysis. The macro and meso scales design variables are simultaneously updated using the Method of Moving Asymptotes (MMA) to find concurrently optimal configurations of constrained and viscoelastic damping layers at the macro scale and viscoelastic damping materials at the micro scale. Two numerical examples are given to demonstrate the effectiveness of the proposed approach.

## 2. Multiscale CLD Structure and Its Damping Property

### 2.1. Effective Properties of Viscoelastic Damping Material

The diagrammatic drawing in Figure 1 illustrates the multiscale CLD structure. The macrostructure, as shown in Figure 1a, whose viscoelastic damping layer is shown in Figure 1b, is represented by the mesostructure shown in Figure 1c. The analysis of the multiscale CLD structure can be divided into two sequential problems. In the first problem (meso scale), the global behavior of a so-called RVE is determined. The effective properties of the viscoelastic damping materials are derived by solving the mesoproblem. These effective properties are then used in the second problem (macro scale) in order to analyze the behavior of the macro scale CLD structure. The reliability of the finite element model of the CLD structure strongly depends on the accuracy of the effective properties of the viscoelastic damping materials. The prediction of the effective properties of the viscoelastic damping materials must be performed as accurately as possible [36]. Due to the warping effect, the deformation of the viscoelastic damping core is complex. For a relatively thin core, the deformation is greatly affected by the skins. Thus, in this paper, the effective constitutive matrix of the viscoelastic damping materials was calculated using the RVE with rigid skin effect [35,37].

As illustrated in Figure 1, the RVE of viscoelastic damping layer is defined as an orthotropic material. The generalized Hooke’s law can be written as follows [38]:(1)$$\left\{\begin{array}{c} {\bar{\sigma }}_{1}\\ {\bar{\sigma }}_{2}\\ {\bar{\sigma }}_{3}\\ {\bar{\sigma }}_{4}\\ {\bar{\sigma }}_{5}\\ {\bar{\sigma }}_{6} \end{array}\right\}=\left[\begin{array}{cccccc} D_{11}^{H} & D_{12}^{H} & D_{13}^{H} & 0 & 0 & 0\\ D_{21}^{H} & D_{22}^{H} & D_{23}^{H} & 0 & 0 & 0\\ D_{31}^{H} & D_{32}^{H} & D_{33}^{H} & 0 & 0 & 0\\ 0 & 0 & 0 & D_{44}^{H} & 0 & 0\\ 0 & 0 & 0 & 0 & D_{55}^{H} & 0\\ 0 & 0 & 0 & 0 & 0 & D_{66}^{H} \end{array}\right]\left\{\begin{array}{c} {\bar{\varepsilon }}_{1}\\ {\bar{\varepsilon }}_{2}\\ {\bar{\varepsilon }}_{3}\\ {\bar{\varepsilon }}_{4}\\ {\bar{\varepsilon }}_{5}\\ {\bar{\varepsilon }}_{6} \end{array}\right\}$$
where $D_{\alpha \beta }^{H}\left(\alpha ,\beta =1,2,\ldots ,6\right)$ represents the components of the effective constitutive matrix $D^{H}$. ${\bar{\sigma }}_{\alpha }$ and ${\bar{\varepsilon }}_{\beta }$ are the volume average stress tensor and volume average strain tensor, respectively.

An eight-node hexahedron element was used to establish the finite element model of the RVE. The nine components of the equivalent elasticity matrix were obtained by solving nine different static analyses for the finite element model of the RVE. Boundary conditions of the RVE can influence the effective constitutive matrix [39]. According to [36,37], the corresponding boundary conditions considering the rigid skin effect for each static analysis are shown in Table 1.

The arbitrarily imposed averaged strains are expressed as follows:(2)$${\bar{\varepsilon }}_{01}=\frac{u}{a},{\bar{\varepsilon }}_{02}=\frac{u}{b},{\bar{\varepsilon }}_{03}=\frac{u}{c},{\bar{\varepsilon }}_{04}=\frac{u}{c},{\bar{\varepsilon }}_{05}=\frac{u}{c},{\bar{\varepsilon }}_{06}=\frac{u}{b}$$
where u is the arbitrarily imposed displacement. a, b and c are the length, width, and height of the RVE, respectively.

The total strain density energy corresponding to the *i*-th load case can be obtained as follows:(3)$$U_{i}=\frac{1}{2V_{\mathrm{RVE}}}\sum_{e=1}^{n}{\left(u_{e}^{i}\right)}^{T}k_{e}u_{e}^{i}=\frac{1}{2V_{\mathrm{RVE}}}\sum_{e=1}^{n}{\left(u_{e}^{i}\right)}^{T}(\int_{Y_{e}}b^{T}D_{e}^{\mathrm{ME}}bdY_{e})u_{e}^{i}$$
where $V_{\mathrm{RVE}}$ is the volume of the RVE; n is the total number of the elements; $k_{e}$ and $D_{e}^{\mathrm{ME}}$ are the stiffness matrix and constitutive matrix of the *e*-th element, respectively; $u_{e}^{i}$ represents the element displacement solutions corresponding to the *i*-th load case, and b is the strain matrix on the meso scale.

From the first six load cases in Table 1, only one component of the strain is non-zero. According to [40], it is obtained as follows:(4)$$D_{ii}^{H}=\frac{2U_{i}}{{\bar{\varepsilon }}_{0i}^{2}};\;i=1,\;2,\;3\cdots ,\;6$$

From the last three load cases in Table 1, for which two components of the strain are non-zero, the results are as follows:(5)$$\begin{array}{l} D_{12}^{H}=D_{21}^{H}=\frac{U_{7}-U_{1}-U_{2}}{{\bar{\varepsilon }}_{01}{\bar{\varepsilon }}_{02}}\\ D_{13}^{H}=D_{31}^{H}=\frac{U_{8}-U_{1}-U_{3}}{{\bar{\varepsilon }}_{01}{\bar{\varepsilon }}_{03}}\\ D_{23}^{H}=D_{32}^{H}=\frac{U_{9}-U_{2}-U_{3}}{{\bar{\varepsilon }}_{02}{\bar{\varepsilon }}_{03}} \end{array}$$

The effective density of the RVE was evaluated through the following relationship
(6)$$\rho^{H}=\frac{\sum_{e=1}^{n}\rho_{e}^{v}}{n}$$
where $\rho_{e}^{v}$ is the density of the *e*-th element.

### 2.2. Damping Property Analysis

It was assumed that the damping characteristic of the viscoelastic damping material is expressed as follows:(7)$$D^{v}=D^{v^{\prime }}+iD^{v^{\prime \prime }}=D^{v^{\prime }}(1+i\eta )$$
where $D^{v^{\prime }}$ and $D^{v^{\prime \prime }}$ are the real part and imaginary part of the constitutive matrix, respectively; η is the material loss factor; and i is the imaginary unit, $i=\sqrt{-1}$.

In the analysis of a structure with the CLD treatment by using the finite element method, the momentum equation for the free vibration of the structure is written as follows:(8)$$M\ddot{X}+(K_{R}+iK_{I})X=0$$
where M is the global mass matrix; $K_{R}$ and $K_{I}$ are the real part and imaginary part of the global stiffness matrix, respectively; and X is the nodal displacement vector.

The global mass and stiffness matrices can be expressed as follows
(9)M=∑i=1Nm(Mib+Miv+Mic)KR=∑i=1Nm(Kib+Re(Kiv)+Kic)KI=∑i=1NmIm(Kiv)
where $K_{i}^{b}$, $K_{i}^{v}$ and $K_{i}^{c}$ are the *i*-th element stiffness matrices; $M_{i}^{b}$, $M_{i}^{v}$ and $M_{i}^{c}$ are the *i*-th element mass matrices; the superscripts b, v and c represent the base structure, the constrained layer and the viscoelastic damping layer, respectively; and ‘Re’ represents for ‘Real part of’, while ‘Im’ denotes the ‘Imaginary part of’. $K_{i}^{v}$ and $M_{i}^{v}$ were determined by using the effective properties of viscoelastic damping material.

According to the Modal Strain Energy (MSE) method, the MLF $\eta_{r}$ is expressed as follows:(10)$$\eta_{r}=\frac{\Phi_{r}^{T}K_{I}\Phi_{r}}{\Phi_{r}^{T}K_{R}\Phi_{r}}$$
where $\Phi_{r}$ is the eigenvector, in which only the real part of the stiffness matrix participates in the modal analysis.

## 3. Multiscale Topology Optimization

### 3.1. Problem Statement and Material Interpolation Scheme

The objective of the CLD treatment is to dissipate the vibrational energy, which can be improved by maximizing the MLF. The concurrent optimization problem regarding the macro and meso scales can be formulated as follows:(11)$$\begin{array}{l} Find:X(\rho_{i}^{MA},\;\rho_{j}^{ME})\\ Minimize:f=\sum_{r=1}^{m}\frac{\alpha_{r}}{\eta_{r}}\\ Subjec\;tto:\;V_{f}^{MA}\geq \frac{1}{V^{MA}}\sum_{i=0}^{Nm}V_{i}\rho_{i}^{MA}\;\\ \;\;\;\;\;V_{f}^{ME}\geq \frac{1}{V^{ME}}\sum_{j=0}^{n}V_{j}\rho_{j}^{ME}\\ \;\;\;\;\;0<\rho_{\min}^{MA}\leq \rho_{i}^{MA}\leq 1,\;i=1,\;2,\;\cdots ,\;Nm\\ \;\;\;\;\;0<\rho_{\min}^{\mathrm{ME}}\leq \rho_{j}^{ME}\leq 1,\;j=1,\;2,\;\cdots ,\;n \end{array}$$
where X consists of the subsets of the design variables for both domains, the macro relative density $\rho_{i}^{MA}$ and the meso relative density $\rho_{j}^{ME}$. $\rho_{i}^{MA}$ describes the layouts of the macro scale of the constrained layer and viscoelastic damping layer, and $\rho_{j}^{ME}$ describes the layouts of the meso scale of viscoelastic damping layer. $\alpha_{r}$ is the weight coefficient. $V_{f}^{MA}$ and $V_{f}^{ME}$ are the volume fractions on the macro and meso scales, respectively.$V_{i}$ is the volume of the element i on the macro scale and $V_{j}$ is the volume of the element j on the meso scale. $\rho_{\min}^{MA}$ and $\rho_{\min}^{ME}$ are the small positive values for the lower bound of the design variables to avoid the singularity of the stiffness matrix. Nm is the total number of the design variables on the macro scale.

In order to achieve a clear design layout, the penalization method was applied. Based on the Rational Approximation of Material Properties (RAMP) model [41], the mass density and elasticity matrix of the *i*-th element in the RVE can be written as follows:(12)$$\begin{array}{l} \rho_{j}^{v}=\rho_{j}^{\mathrm{ME}}\rho^{v}\\ D_{j}^{\mathrm{ME}}=\frac{\rho_{j}^{ME}}{1+p(1-\rho_{j}^{ME})}D^{v} \end{array}$$
where $\rho_{j}^{v}$ and $D_{j}^{ME}$ are the mass density and the elasticity matrix, respectively, of the *j*-th element in the RVE. $\rho^{v}$ and $D^{v}$ are the mass density and the elasticity matrix, respectively, of the viscoelastic damping material. p is the exponent of penalization.

On the macro scale, the design region is composed of a constrained layer and a viscoelastic damping layer. Using the same interpolation scheme, the mass and stiffness matrices can be interpolated as follows:(13)$$\begin{array}{l} M_{i}^{v}=\rho_{i}^{MA}\int_{\Omega_{i}}\rho^{H}N^{T}Nd\Omega_{i}\\ M_{i}^{c}=\rho_{i}^{MA}\int_{\Omega_{i}}\rho^{c}N^{T}Nd\Omega_{i}\\ K_{i}^{v}=\frac{\rho_{i}^{\mathrm{MA}}}{1+p(1-\rho_{i}^{\mathrm{MA}})}\int_{\Omega_{i}}B^{T}D^{H}Bd\Omega_{i}\\ K_{i}^{c}=\frac{\rho_{i}^{\mathrm{MA}}}{1+p(1-\rho_{i}^{\mathrm{MA}})}\int_{\Omega_{i}}B^{T}D^{c}Bd\Omega_{i} \end{array}$$
where $D^{b}$ and $D^{c}$ are constitutive matrices; $\rho^{b}$ and $\rho^{c}$ represent mass density; and B and N are the strain matrix and the shape function matrix on the macro scale, respectively.

### 3.2. Sensitivity Analysis on the Macro Scale

The sensitivity of the objective function in Equation (11) with respect to the design variables in the macrostructure can be expressed as follows:(14)$$\frac{\partial f}{\partial \rho_{i}^{MA}}=\sum_{r=1}^{m}\alpha_{r}\frac{\partial }{\partial \rho_{i}^{MA}}(\frac{1}{\eta_{r}})$$

The sensitivity of the objective function with respect to the design variables $\rho_{i}^{MA}$ was determined by considering the contribution of the sensitivity of the eigenvectors with respect to the design. The sensitivity analysis scheme for the reciprocal of MLF was derived by using the adjoint variable method (AVM), which eliminated the unknown expression involving the sensitivities of the eigenvectors and eigenvalues with respect to the design [42]. The adjoint variables $\mu_{1}$ and $\mu_{2}$ were introduced and the reciprocal of MLF $1/\eta_{r}$ was rewritten as follows:(15)$$\frac{1}{\eta_{r}}=\frac{\Phi_{r}^{T}K_{R}\Phi_{r}}{\Phi_{r}^{T}K_{I}\Phi_{r}}+\mu_{1}^{T}(K_{R}-\lambda_{r}^{2}M)\Phi_{r}+\mu_{2}(\Phi_{r}^{T}M\Phi_{r}-1)$$
where $\lambda_{r}^{2}$ is the eigenvalue at the *r*-th mode.

The sensitivity of $1/\eta_{r}$ with respect to the design variables in the macrostructure was taken from Equation (15) as follows:(16)$$\begin{array}{ll} \frac{\partial }{\partial \rho_{i}^{MA}}(\frac{1}{\eta_{r}}) & =(\frac{\left(\Phi_{r}^{T}K_{I}\Phi_{r})(2\Phi_{r}^{T}K_{R})-(\Phi_{r}^{T}K_{R}\Phi_{r})(2\Phi_{r}^{T}K_{I}\right)}{{\left(\Phi_{r}^{T}K_{I}\Phi_{r}\right)}^{2}}+\mu_{1}^{T}(K_{R}-\lambda_{r}^{2}M)+2\mu_{2}\Phi_{r}^{T}M)\frac{\partial \Phi_{r}}{\partial \rho_{i}^{MA}}\\ & +\frac{\left(\Phi_{r}^{T}K_{I}\Phi_{r})(\Phi_{r}^{T}\frac{\partial K_{R}}{\partial \rho_{i}^{\mathrm{MA}}}\Phi_{r})-(\Phi_{r}^{T}K_{R}\Phi_{r})(\Phi_{r}^{T}\frac{\partial K_{I}}{\partial \rho_{i}^{\mathrm{MA}}}\Phi_{r}\right)}{{\left(\Phi_{r}^{T}K_{I}\Phi_{r}\right)}^{2}}\\ & \;+\mu_{1}^{T}(\frac{\partial K_{R}}{\partial \rho_{i}^{MA}}-\lambda_{r}^{2}\frac{\partial M}{\partial \rho_{i}^{MA}})\Phi_{r}-\mu_{1}^{T}\frac{\partial \lambda_{r}^{2}}{\partial \rho_{i}^{MA}}M\Phi_{r}+\mu_{2}\Phi_{r}^{T}\frac{\partial M}{\partial \rho_{i}^{MA}}\Phi_{r} \end{array}$$

To remove the implicit derivatives of the eigenvectors and eigenvalues, the adjoint variables should satisfy the following equations
(17)$$\begin{array}{l} \frac{\left(\Phi_{r}^{T}K_{I}\Phi_{r})(2\Phi_{r}^{T}K_{R})-(\Phi_{r}^{T}K_{R}\Phi_{r})(2\Phi_{r}^{T}K_{I}\right)}{{\left(\Phi_{r}^{T}K_{I}\Phi_{r}\right)}^{2}}+\mu_{1}^{T}(K_{R}-\lambda_{r}^{2}M)+2\mu_{2}\Phi_{r}^{T}M=0\\ \mu_{1}^{T}M\Phi_{r}=0 \end{array}$$

The adjoint variables $\mu_{1}$ and $\mu_{2}$ were found by solving Equation (17). The sensitivity of $1/\eta_{r}$ with respect to the design variables on the macro scale was reformulated from Equation (16) as follows:(18)$$\begin{array}{ll} \frac{\partial }{\partial \rho_{i}^{\mathrm{MA}}}(\frac{1}{\eta_{r}}) & =\frac{\left(\Phi_{r}^{T}K_{I}\Phi_{r})(\Phi_{r}^{T}\frac{\partial K_{R}}{\partial \rho_{i}^{\mathrm{MA}}}\Phi_{r})-(\Phi_{r}^{T}K_{R}\Phi_{r})(\Phi_{r}^{T}\frac{\partial K_{I}}{\partial \rho_{i}^{\mathrm{MA}}}\Phi_{r}\right)}{{\left(\Phi_{r}^{T}K_{I}\Phi_{r}\right)}^{2}}\\ & +\mu_{1}^{T}(\frac{\partial K_{R}}{\partial \rho_{i}^{\mathrm{MA}}}-\lambda_{r}^{2}\frac{\partial M}{\partial \rho_{i}^{\mathrm{MA}}})\Phi_{r}+\mu_{2}\Phi_{r}^{T}\frac{\partial M}{\partial \rho_{i}^{\mathrm{MA}}}\Phi_{r} \end{array}$$

The derivatives of the global stiffness and mass matrices with respect to the design variables $\rho_{i}^{MA}$ can be expressed as follows:(19)∂KR∂ρiMA=1+p(1+p(1−ρiMA))2(Re(∫ΩiBTDHBdΩi)+∫ΩiBTDcBdΩi)∂KI∂ρiMA=1+p(1+p(1−ρiMA))2Im(∫ΩiBTDHBdΩi)∂M∂ρiMA=∫ΩiρHNTNdΩi+∫ΩiρcNTNdΩi

### 3.3. Sensitivity Analysis on the Meso scale

For sensitivity analysis on the meso scale, the sensitivity of the eigenvectors with respect to design variables $\rho_{j}^{ME}$ was ignored to improve the computational efficiency. The sensitivity of $1/\eta_{r}$ with respect to the design variables in the mesostructure was taken from Equation (10) as follows:(20)$$\frac{\partial }{\partial \rho_{j}^{\mathrm{ME}}}(\frac{1}{\eta_{r}})=\frac{\left(\Phi_{r}^{T}K_{I}\Phi_{r})(\Phi_{r}^{T}\frac{\partial K_{R}}{\partial \rho_{j}^{ME}}\Phi_{r})-(\Phi_{r}^{T}K_{R}\Phi_{r})(\Phi_{r}^{T}\frac{\partial K_{I}}{\partial \rho_{j}^{ME}}\Phi_{r}\right)}{{\left(\Phi_{r}^{T}K_{I}\Phi_{r}\right)}^{2}}$$

The sensitivities of the global stiffness matrices with respect to the design variables $\rho_{j}^{ME}$ can be derived as follows:(21)∂KR∂ρjME=∑i=1NmρiMA1+p(1−ρiMA)∫ΩiBTRe(∂DHρjME)BdΩi∂KI∂ρjME=∑i=1NmρiMA1+p(1−ρiMA)∫ΩiBTIm(∂DHρjM)BdΩi

Considering the components of the equivalent constitutive matrix expressed as Equations (4) and (5), the total strain density energy defined in Equation (3) and the material interpolation proposed in Equation (12), the sensitivity of the components of the equivalent constitutive matrix with respect to the design variables on the meso scale can be formulated as follows
(22)$$\frac{\partial D_{ii}^{H}}{\partial \rho_{j}^{\mathrm{ME}}}=\frac{\frac{1+p}{{\left(1+p(1-\rho_{j}^{\mathrm{ME}})\right)}^{2}}{\left(u_{j}^{i}\right)}^{T}(\int_{Y_{j}}b^{T}D^{v}bdY_{j})u_{j}^{i}}{V_{RVE}{\bar{\varepsilon }}_{0i}^{2}};\;i=1,\;2,\;3\cdots ,\;6$$
(23)$$\begin{array}{l} \frac{\partial D_{12}^{H}}{\partial \rho_{j}^{\mathrm{ME}}}=\frac{\partial D_{21}^{H}}{\partial \rho_{j}^{\mathrm{ME}}}=\frac{\frac{1+p}{{\left(1+p(1-\rho_{j}^{\mathrm{ME}})\right)}^{2}}({\left(u_{j}^{7}\right)}^{T}(\int_{Y_{j}}b^{T}D^{v}bdY_{j})u_{j}^{7}-{\left(u_{j}^{1}\right)}^{T}(\int_{Y_{j}}b^{T}D^{v}bdY_{j})u_{j}^{1}-{\left(u_{j}^{2}\right)}^{T}(\int_{Y_{j}}b^{T}D^{v}bdY_{j})u_{j}^{2})}{2V_{RVE}{\bar{\varepsilon }}_{01}{\bar{\varepsilon }}_{02}}\\ \frac{\partial D_{13}^{H}}{\partial \rho_{j}^{\mathrm{ME}}}=\frac{\partial D_{31}^{H}}{\partial \rho_{j}^{\mathrm{ME}}}=\frac{\frac{1+p}{{\left(1+p(1-\rho_{j}^{\mathrm{ME}})\right)}^{2}}({\left(u_{j}^{8}\right)}^{T}(\int_{Y_{j}}b^{T}D^{v}bdY_{j})u_{j}^{8}-{\left(u_{j}^{1}\right)}^{T}(\int_{Y_{j}}b^{T}D^{v}bdY_{j})u_{j}^{1}-{\left(u_{j}^{3}\right)}^{T}(\int_{Y_{j}}b^{T}D^{v}bdY_{j})u_{j}^{3})}{2V_{RVE}{\bar{\varepsilon }}_{01}{\bar{\varepsilon }}_{03}}\\ \frac{\partial D_{23}^{H}}{\partial \rho_{j}^{\mathrm{ME}}}=\frac{\partial D_{32}^{H}}{\partial \rho_{j}^{\mathrm{ME}}}=\frac{\frac{1+p}{{\left(1+p(1-\rho_{j}^{\mathrm{ME}})\right)}^{2}}({\left(u_{j}^{9}\right)}^{T}(\int_{Y_{j}}b^{T}D^{v}bdY_{j})u_{j}^{9}-{\left(u_{j}^{2}\right)}^{T}(\int_{Y_{j}}b^{T}D^{v}bdY_{j})u_{j}^{2}-{\left(u_{j}^{3}\right)}^{T}(\int_{Y_{j}}b^{T}D^{v}bdY_{j})u_{j}^{3})}{2V_{RVE}{\bar{\varepsilon }}_{02}{\bar{\varepsilon }}_{03}} \end{array}$$

### 3.4. Optimization Procedure

Once the sensitivity information was obtained, the MMA was used to update the design variables on both the macro and meso scales. The proposed concurrent topology optimization process for maximizing the MLF of PCLD structures is shown in Figure 2. The main steps are summarized as follows:
(1)Define the design domain and initialize the design variables on both scales;(2)Establish the finite element model of the RVE and calculate the effective complex matrix and density of the viscoelastic damping materials by using the RVE with a rigid skin effect;(3)Establish the finite element model on the macro scale and obtain the MLFs based on the Modal Strain Energy Method;(4)Calculate the sensitivities of the objective function with respect to the design variables on both the macro and meso scales. To circumvent the checkerboard and mesh-dependency problems, a mesh-independence filter scheme [43] was employed to smooth the element sensitivities;(5)Update the design variables on both scales by using MMA; and(6)Check the convergence of the result. If the change in the objective function of twenty successive iterations is less than $10^{-3}$, or the number of iterations reaches the preset iteration number $N_{t}$, the iteration process terminates; otherwise, steps 2–6 are repeated.

## 4. Numerical Examples

### 4.1. The Effective Material Property Analysis

The unit cell of the viscoelastic damping material is shown in Figure 3, which is a cuboid with a square through hole in the center. The geometry of the unit cell was considered as follows: l=0.4 mm; $l_{1}=0.3\;\mathrm{mm}$; cell height variable from h=0.2 mm to h=4 mm. The real part of the complex elastic modulus and Poisson’s ratio of the viscoelastic damping material were 20 MPa and 0.495. The equivalent constitutive matrices were obtained by using the RVE with a rigid skin effect and the classical homogenization method in [20]. The CLD structures dissipate vibration energy through transverse shear strains induced in the viscoelastic damping layer, so the effective transverse shear modulus was the main focus. For the unit cell shown in Figure 3, the effective transverse shear moduli $D_{44}^{H}$ and $D_{55}^{H}$ are the same [44], and Figure 4 shows the comparison of the real part of the effective transverse shear modulus. It can be seen that the real part of the effective transverse shear modulus obtained by the RVE with a rigid skin effect converge to the constant obtained using the classical homogenization method when h tended to infinity. However, the result calculated using the RVE with rigid skin effect was obviously bigger than that calculated using the classical homogenization method when h/l=1; this is because, in contrast to the deformation of the free material, the deformation of the core in the sandwich structure close to the skins follow the skins deformation [36]. Hence, the RVE with rigid skin effect was used to calculate the effective material properties of the viscoelastic damping material in the CLD structure to improve the calculation accuracy.

### 4.2. A rectangular Plate with Four Edges Clamped

A rectangular plate with four edges clamped is shown in Figure 5. The length and width of the rectangular plate were 0.4 and 0.3 m, respectively. The thickness of the base plate, viscoelastic damping layer and constrained layer were 0.001, 0.0004 and 0.0001 m, respectively. The material properties are documented in Table 2. The viscoelastic damping layer consisted of a periodic material, which is represented by the RVE. The size of the RVE was 0.4×0.4×0.4 mm. A mesh with 20×20×20 elements was applied to discretize the RVE. The macrostructure of the CLD plate was discretized with 32×24 elements. All of the above parameters also apply to Section 4.3.

The initial designs for RVE had a uniform distribution with a given volume fraction $V_{f}^{ME}$, except the elements at the center or the corners. Figure 6 shows the initial designs of the RVE, in which the first initial guess shown in Figure 6a is that the density of the mesodesign variables at the center is set to 1, while the second initial guess shown in Figure 6b is that the density of the mesodesign variables at the corners is set to 1. It means that all the parameters of initial guess design 1 and initial guess design 2 were the same except the mesodesign variables of the elements at the center or corner. The initial designs of the macrostructure had a uniform distribution with a given volume fraction $V_{f}^{MA}$.

#### 4.2.1. The Effect of Penalty Factor and Initial Guess Design on the Optimal Topologies

The penalty factor p in the RAMP model had a great influence on the optimization result. Therefore, it was necessary to select the appropriate penalty factor. The design objective is to maximize the MLF of Mode 1 (Objective 1). The volume constraints are $V_{f}^{\mathrm{MA}}=0.8$ and $V_{f}^{\mathrm{ME}}=0.4$. A filter radius of 1.5 times the element size was applied on two scales. The initial design of the RVE is shown in Figure 6a. By choosing five different p values, the optimization results, iteration histories and the first MLF of the optimization results are, respectively, shown in Figure 7, Figure 8 and Figure 9. On the macro scale, the black domain represents CLD material on the base plate. On the meso scale, the red elements represent the viscoelastic damping material, and the brown elements denote intermediate density elements. From Figure 9, it is not difficult to see that the MLFs decreased with the increase of penalty factor. When p=0, almost all elements in the mesostructure were intermediate density elements. When p=1, the mesostructure had many intermediate density elements and the optimization efficiency was the lowest. Therefore, p was set to 2 in the following examples.

Figure 10 shows the optimization results designed from the initial guess design 2 (shown in Figure 6b), and Figure 11 shows the iteration histories. Compared with Figure 7c, as argued in [45], different initial guess designs may lead to different mesostructures and equivalent constitutive matrices. However, the optimized macrostructures were almost unchanged and the design objective of optimized structure is 0.240. Therefore, in the following sections, all optimized structures were designed from initial guess design 1.

#### 4.2.2. Different Optimization Objectives

Two other optimization objectives are discussed here, namely, maximizing the second MLF (Objective 2) and maximizing the sum of the first two MLFs (Objective 3). The volume fractions were also $V_{f}^{\mathrm{MA}}=0.8$ and $V_{f}^{\mathrm{ME}}=0.4$. The optimization results and iteration histories are shown in Figure 12 and Figure 13, respectively. From Figure 7c and Figure 12, it can be seen that the optimized layouts in both scales were different with variation in the optimization objective, and the equivalent constitutive matrices of the first three mesostructures were obviously different. As shown in Figure 8c and Figure 13, the MLF generally increased during the optimization process, and the MLF of the optimized structures were much bigger than those of initial structures. Table 3 is the comparison of the MLFs between optimized structure and full coverage structure. In Table 3, the optimized layouts of Objective 1 are shown in Figure 7c. For the single mode design, the MLFs of optimal designs were increased by 4.35% and 8.33% compared to the full coverage structure in Modes 1 and 2, respectively. Considering the sum of the first two MLFs, the MLFs of optimal design were increased by 4.26% compared to that of the traditional design. When maximization of the first MLF was the design objective (Objective 1), the first MLF of the optimized structure was the maximum. Objective 2 and 3 have the same tendency with Objective 1.

A harmonic excitation force f with the amplitude F = 1 N was applied to the rectangular plate with four edges clamped. The response point and the excitation point were in the same place. A comparison of the frequency response curves of the full coverage structure and optimal designs is shown in Figure 14, in which Figure 14b,c shows a detailed description in Modes 1 and 2. It can be seen that Objective 1 had the smallest resonance peak at the first eigenmode and Objective 2 had the smallest resonance peak at the second eigenmode, while Objective 3 obtained a better equilibrium in the first two modes. Compared with the full coverage structure, the optimized structures still have good vibration characteristics while reducing the consumption of the CLD material.

#### 4.2.3. Different Volume Fraction

In this section, the effect of the volume fraction on the optimization results is investigated. Two optimization objectives were maximizing the first MLF and the second MLF, respectively. For each optimization objective, three combinations of $V_{f}^{\mathrm{MA}}$ and $V_{f}^{\mathrm{ME}}$ were tested. The optimum designs are shown in Figure 15 and Figure 16. Iteration histories are shown in Figure 17 and Figure 18. The MLFs of the optimum designs are shown in Table 4. As shown in Figure 15 and Figure 16, the optimized layouts of the macrostructure were not only affected by the objective mode but also by the volume fraction on the meso scale. The optimized designs on the meso scale are mainly related to the objective mode. When varying the volume fraction on the macro scale, the optimized designs on the meso scale were different only in terms of their detailed sizes, which are reflected in the values of the equivalent constitutive matrices. From Figure 17 and Figure 18 and Table 4, it can be seen that the MLF generally increased during the optimization process, and the MLFs of the optimized structures were much bigger than those of the initial structures.

### 4.3. A Rectangular Plate with Two Short Edges Clamped

#### 4.3.1. Different Optimization Objectives

In order to further verify the effectiveness of the proposed concurrent topology optimization method, another rectangular plate with a different boundary condition is studied in this section. The rectangular plate with two short edges clamped is shown in Figure 19. Three optimization objectives are discussed here, namely, maximizing the first MLF (Objective 1), maximizing the second MLF (Objective 2) and the sum of the first two MLFs (Objective 3). The volume fractions are the same as the cases in Section 4.2.1. A filter radius of 4 times the element size was used on the macro scale. The filter radius on the meso scale was 1.5 times the element size. The design results and iteration histories are shown in Figure 20 and Figure 21, respectively. Figure 22a shows the comparison of the frequency response curves of full coverage structure and optimal designs, and Figure 22b,c provides a detailed description of Modes 1 and 2. From Figure 20, the similar conclusions can be obtained that the optimized designs in both scales are dependent on the objective mode. Therefore, the equivalent constitutive matrices of the three designs on the meso scale are obviously different. As shown in Figure 21, the MLF generally increased during the optimization process, and the MLFs of the optimized structures were much bigger than those of initial structures. The comparison of the MLFs between the design results and full coverage structure is shown in Table 5. For the single mode design, the MLFs of optimal designs were increased by 17.39% and 31.58% compared to the full coverage structure in Modes 1 and 2, respectively. Considering the sum of the first two MLFs, the MLFs of optimal design was increased by 19.05% compared to those of the traditional design. From Table 5 and Figure 22, it can be seen that when maximizing the *k*-th (*k* = 1, 2) MLF, the *k*-th MLF is maximum and the resonance peak at the *k*-th eigenmode was the minimum. In the case of maximizing the sum of the first two MLFs, though the MLF and resonance peak in the *k*-th eigenmode showed inferior results which maximized the *k*-th MLF, the optimized design obtained a better equilibrium in the first two modes.

#### 4.3.2. Different Volume Fraction

In this section, the optimization objectives and volume fractions were the same as the cases in Section 4.2.3 and the filter radii in both scales were the same as the cases in Section 4.3.1. The optimum designs are shown in Figure 23 and Figure 24. Iteration histories are shown in Figure 25 and Figure 26. The MLFs of the optimum designs are shown in Table 6. From the above figures and table, similar conclusions can be obtained as in Section 4.2.3.

## 5. Conclusions

A concurrent topology optimization approach is proposed here for the multi-scale design of a CLD plate to maximize the MLF. The equivalent constitutive matrix of viscoelastic damping material was calculated using the RVE with a rigid skin effect and was taken into account in the finite element analysis of the macrostructure of the CLD plate. The sensitivity calculation was performed on both macro and meso scales. The MMA was used to update the design variables on two scales and the optimized design was obtained. The numerical examples are presented using the proposed concurrent topology optimization approach to multiscale systems.

By analyzing the influence of the penalty factor in the RAMP model on optimization results, an appropriate penalty factor was chosen. The effects of optimization objectives and volume fractions on the design results were investigated. The results indicated that the optimized layouts of the macrostructure were dependent on the objective mode and volume fraction on the meso scale. The optimized designs on the meso scale were mainly related to the objective mode. When varying the volume fraction on the macro scale, the optimized designs on the meso scale were different only in their detailed size, which were reflected in the values of the equivalent constitutive matrices. When maximizing the *k*-th MLF, the *k*-th MLF is maximum and the resonance peak at the *k*-th eigenmode was the minimum. In the case of maximizing the sum of the first two MLFs, the optimized design obtained a better equilibrium in the first two modes. The proposed concurrent topology optimization method can provide an effective means to optimize the structural damping of the CLD structure and produce optimal layouts on both the macro and meso scales. The proposed concurrent topology optimization method is a good choice for the optimization of the structural damping of CLD structure and producing optimal layout on both scales.

## Figures and Tables

**Figure 1 materials-15-03512-f001:**
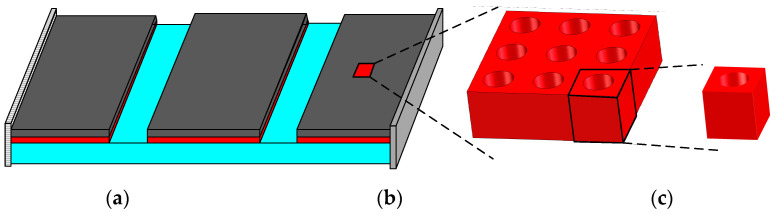
A multiscale system of structure with constrained layer damping treatment: (**a**) Macrostructure; (**b**) periodic material; (**c**) mesostructure.

**Figure 2 materials-15-03512-f002:**
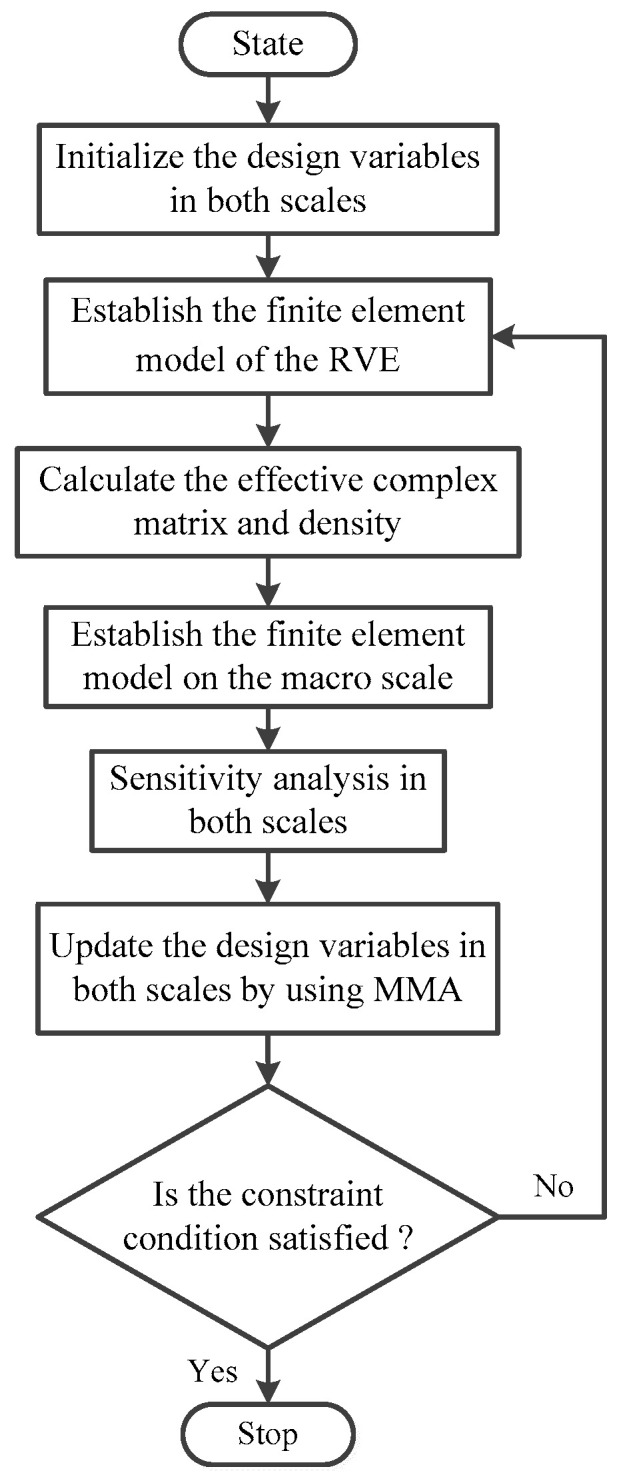
Flowchart of concurrent topology optimization for maximizing the modal loss factor.

**Figure 3 materials-15-03512-f003:**
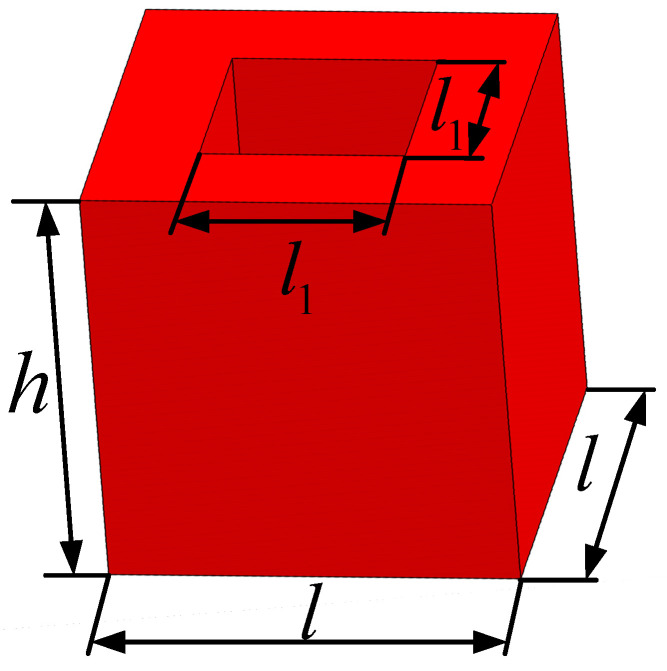
The unit cell.

**Figure 4 materials-15-03512-f004:**
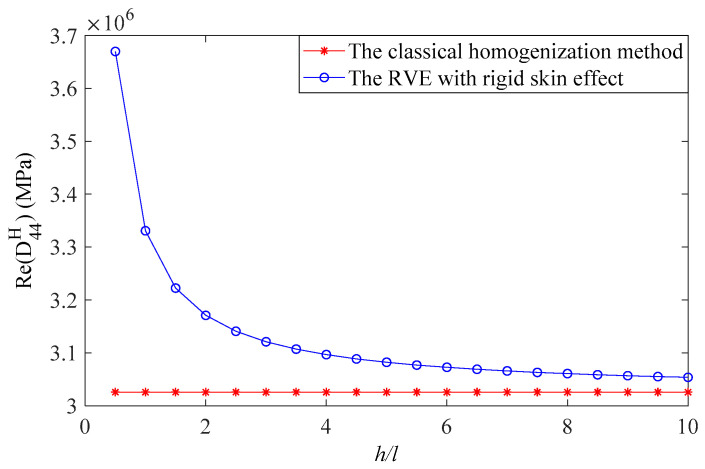
Comparison of the real part of effective transverse shear modulus.

**Figure 5 materials-15-03512-f005:**
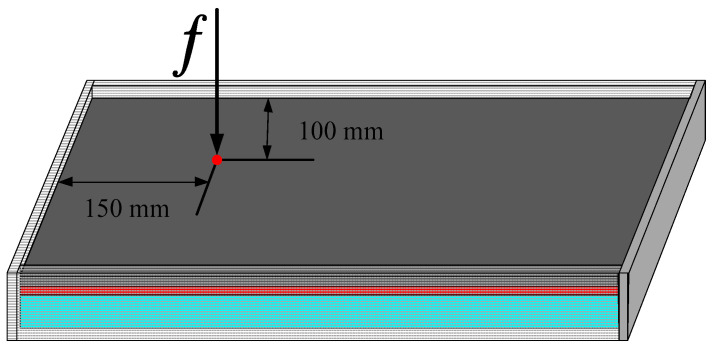
A rectangular plate with four edges clamped.

**Figure 6 materials-15-03512-f006:**
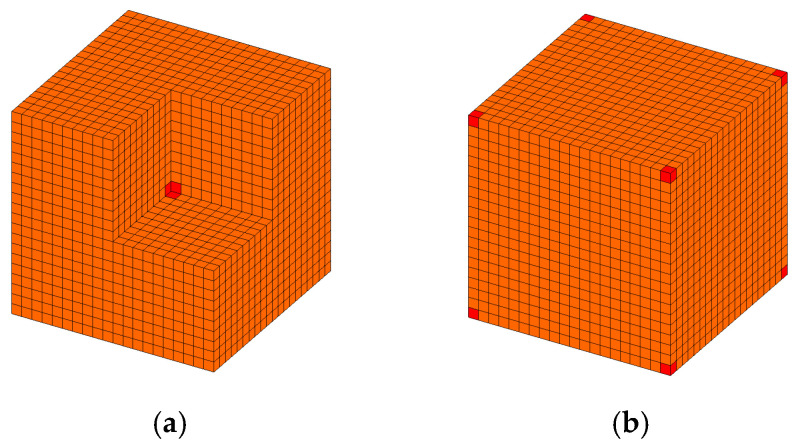
Initial guess design of the representative volume element: (**a**) initial guess design 1 (hide the upper right front part); (**b**) initial guess design 2.

**Figure 7 materials-15-03512-f007:**
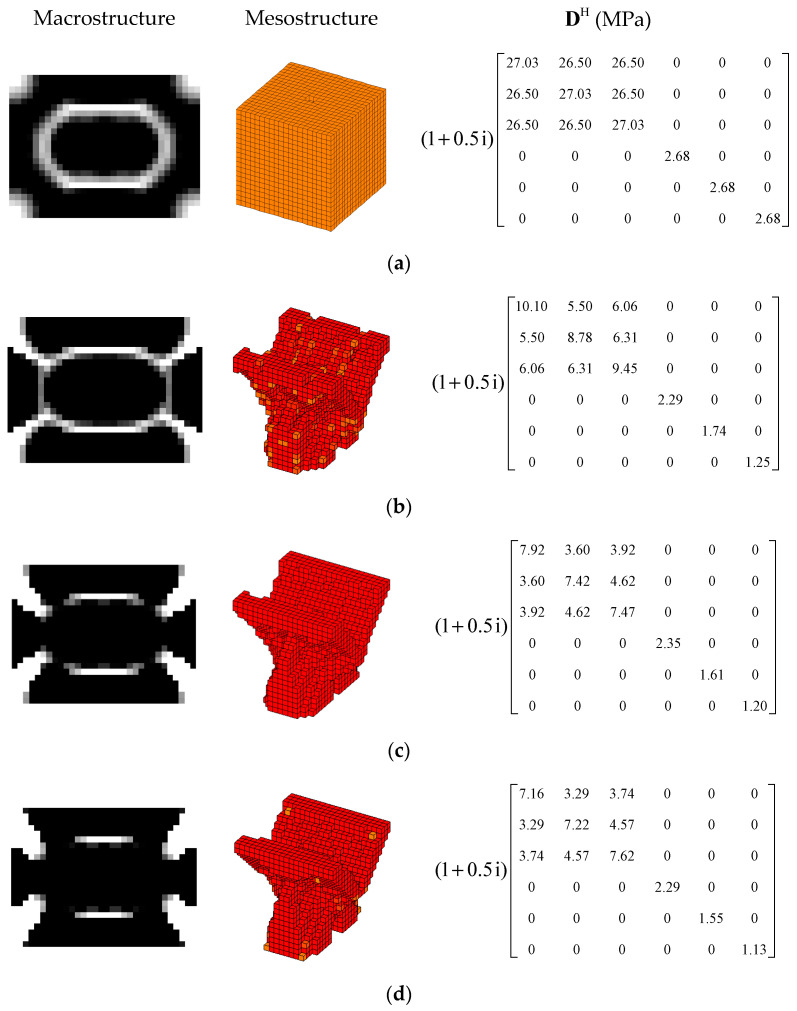
Optimization results obtained by varying the penalty factor: (**a**) p=0; (**b**) p=1; (**c**) p=2; (**d**) p=3; (**e**) p=5.

**Figure 8 materials-15-03512-f008:**
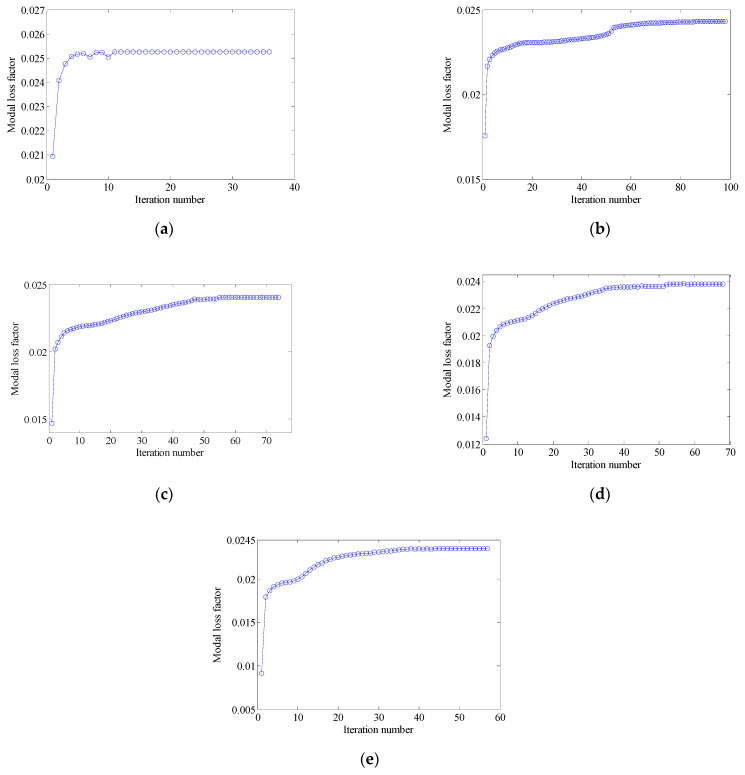
Iteration histories for different penalty factors: (**a**) p=0; (**b**) p=1; (**c**) p=2; (**d**) p=3; (**e**) p=5.

**Figure 9 materials-15-03512-f009:**
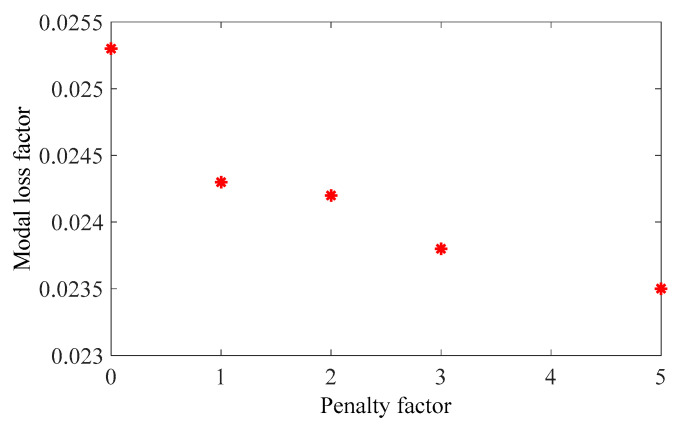
The first modal loss factor of the optimization results obtained by varying the penalty factor.

**Figure 10 materials-15-03512-f010:**
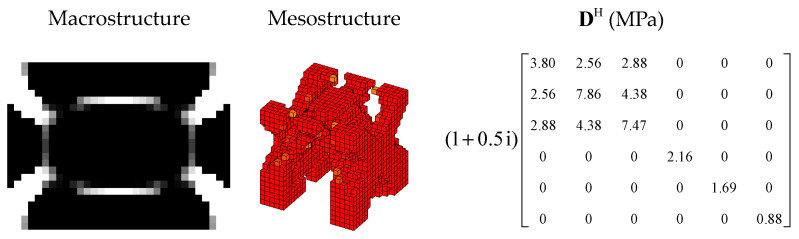
Optimization results designed from initial guess design 2.

**Figure 11 materials-15-03512-f011:**
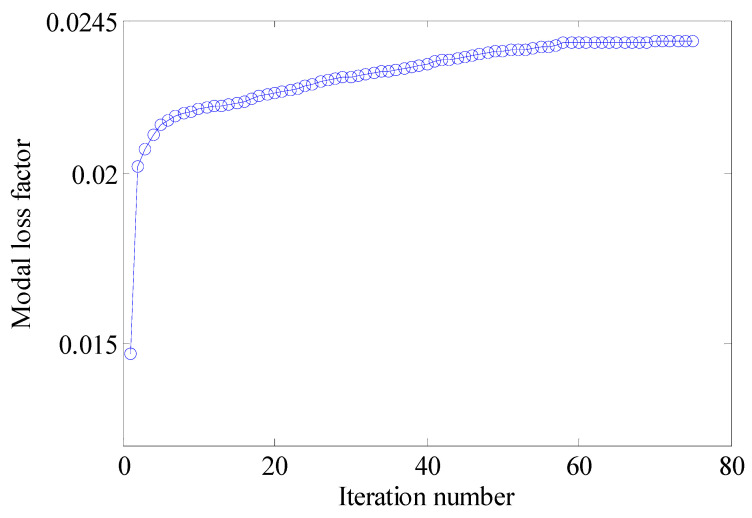
Iteration histories for initial guess design 2.

**Figure 12 materials-15-03512-f012:**
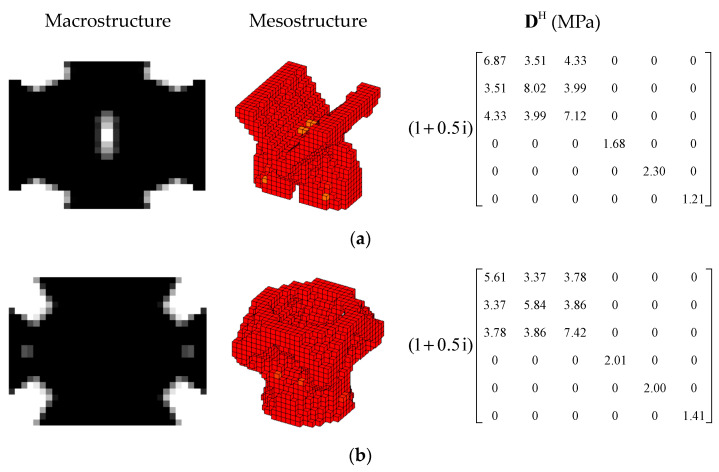
Design results for different optimization objectives: (**a**) Objective 2: maximizing the second modal loss factor; (**b**) Objective 3: Maximizing the sum of the first two modal loss factors.

**Figure 13 materials-15-03512-f013:**
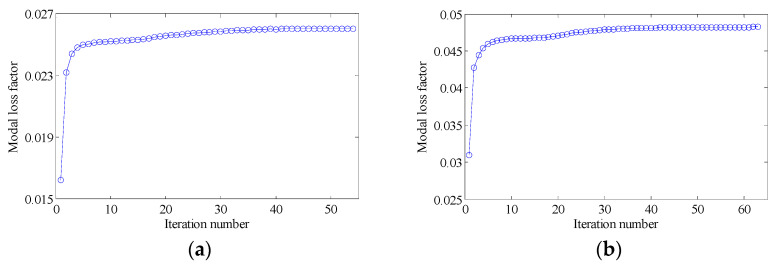
Iteration histories for different optimization objectives: (**a**) Objective 2: maximizing the second modal loss factor; (**b**) Objective 3: Maximizing the sum of the first two modal loss factors.

**Figure 14 materials-15-03512-f014:**
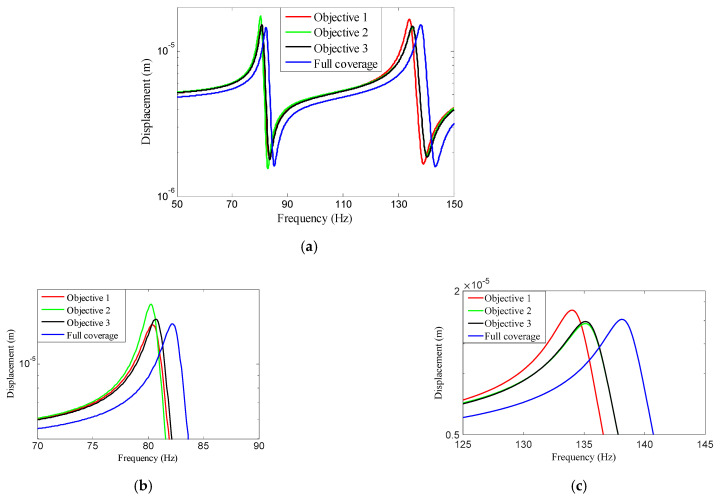
Comparison of the frequency response curves of full coverage structure and optimal designs: (**a**) The frequency response curves from 50 Hz to 150 Hz; (**b**) the detailed description in Mode 1; (**c**) the detailed description in Mode 2.

**Figure 15 materials-15-03512-f015:**
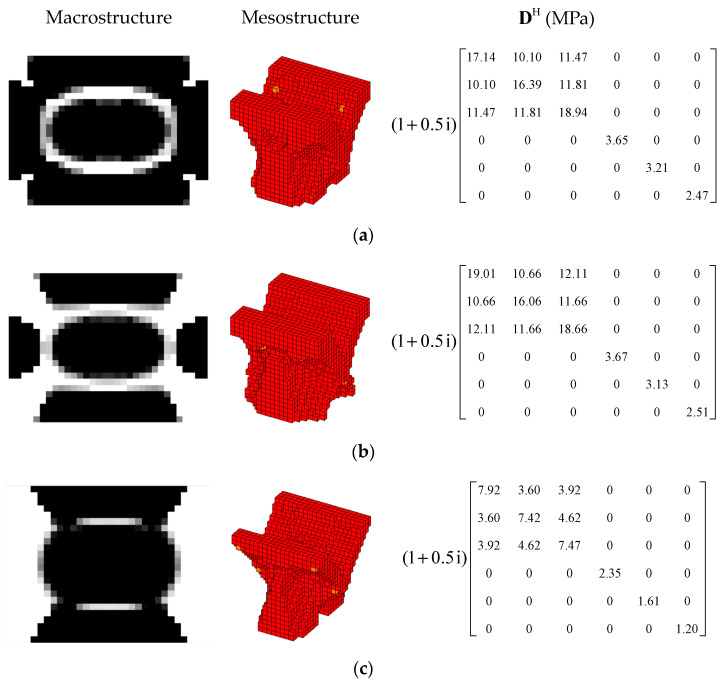
Design results for maximizing the first modal loss factor: (**a**) Case 1: $V_{f}^{\mathrm{MA}}=0.8$ and $V_{f}^{\mathrm{MI}}=0.6$; (**b**) Case 2: $V_{f}^{\mathrm{MA}}=0.6$ and $V_{f}^{\mathrm{MI}}=0.6$; (**c**) Case 3: $V_{f}^{\mathrm{MA}}=0.6$ and $V_{f}^{\mathrm{MI}}=0.4$.

**Figure 16 materials-15-03512-f016:**
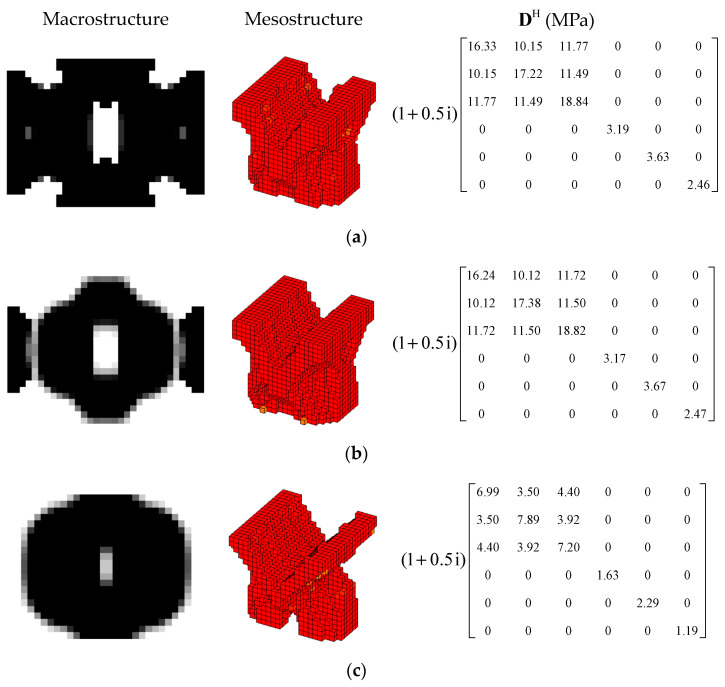
Design results for maximizing the second modal loss factor: (**a**) Case 4: $V_{f}^{\mathrm{MA}}=0.8$ and $V_{f}^{\mathrm{MI}}=0.6$; (**b**) Case 5: $V_{f}^{\mathrm{MA}}=0.6$ and $V_{f}^{\mathrm{MI}}=0.6$; (**c**) Case 6: $V_{f}^{\mathrm{MA}}=0.6$ and $V_{f}^{\mathrm{MI}}=0.4$.

**Figure 17 materials-15-03512-f017:**
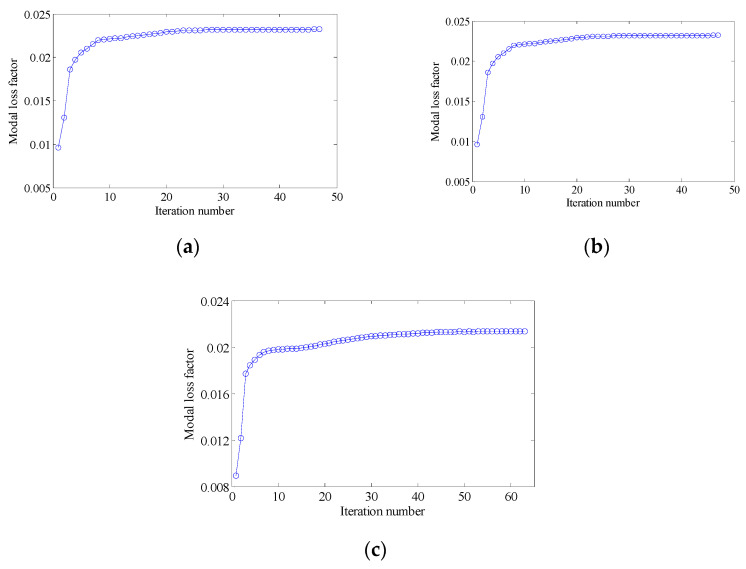
Iteration histories for maximizing the first modal loss factor: (**a**) Case 1: $V_{f}^{\mathrm{MA}}=0.8$ and $V_{f}^{\mathrm{MI}}=0.6$; (**b**) Case 2: $V_{f}^{\mathrm{MA}}=0.6$ and $V_{f}^{\mathrm{MI}}=0.6$; (**c**) Case 3: $V_{f}^{\mathrm{MA}}=0.6$ and $V_{f}^{\mathrm{MI}}=0.4$.

**Figure 18 materials-15-03512-f018:**
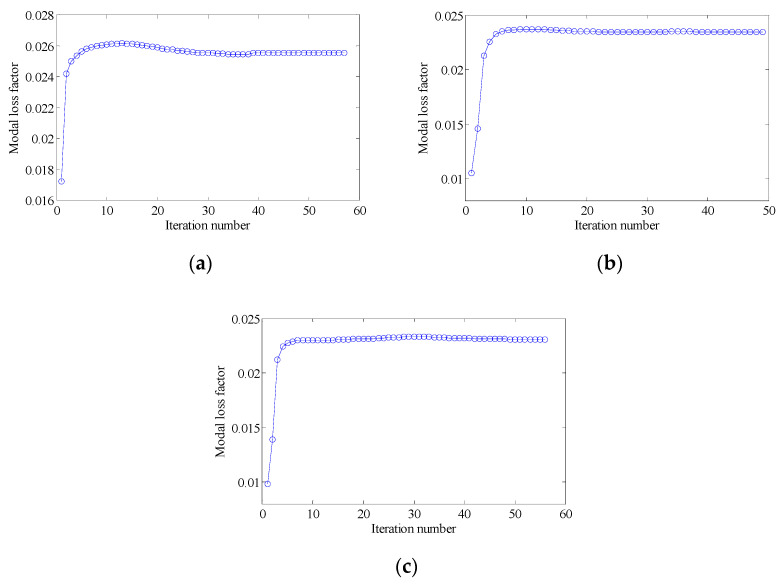
Iteration histories for maximizing the second modal loss factor: (**a**) Case 4: $V_{f}^{\mathrm{MA}}=0.8$ and $V_{f}^{\mathrm{MI}}=0.6$; (**b**) Case 5: $V_{f}^{\mathrm{MA}}=0.6$ and $V_{f}^{\mathrm{MI}}=0.6$; (**c**) Case 6: $V_{f}^{\mathrm{MA}}=0.6$ and $V_{f}^{\mathrm{MI}}=0.4$.

**Figure 19 materials-15-03512-f019:**
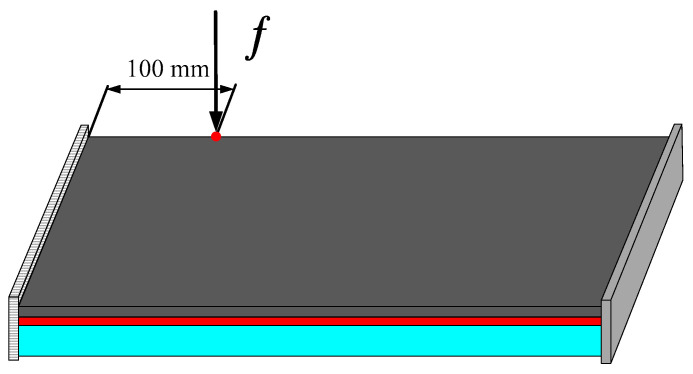
A rectangular plate with two short edges clamped.

**Figure 20 materials-15-03512-f020:**
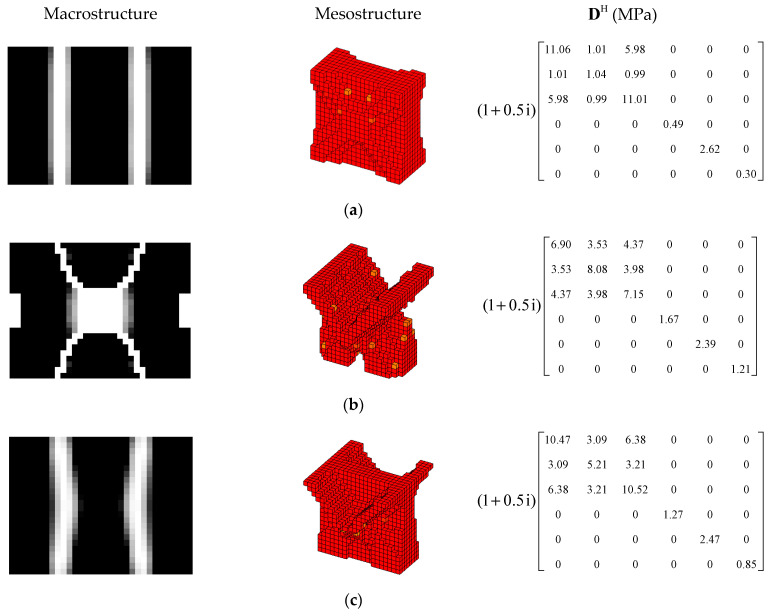
Design results for different optimization objectives: (**a**) Objective 1: maximizing the first modal loss factor; (**b**) Objective 2: maximizing the second modal loss factor; (**c**) Objective 3: maximizing the sum of the first two modal loss factors.

**Figure 21 materials-15-03512-f021:**
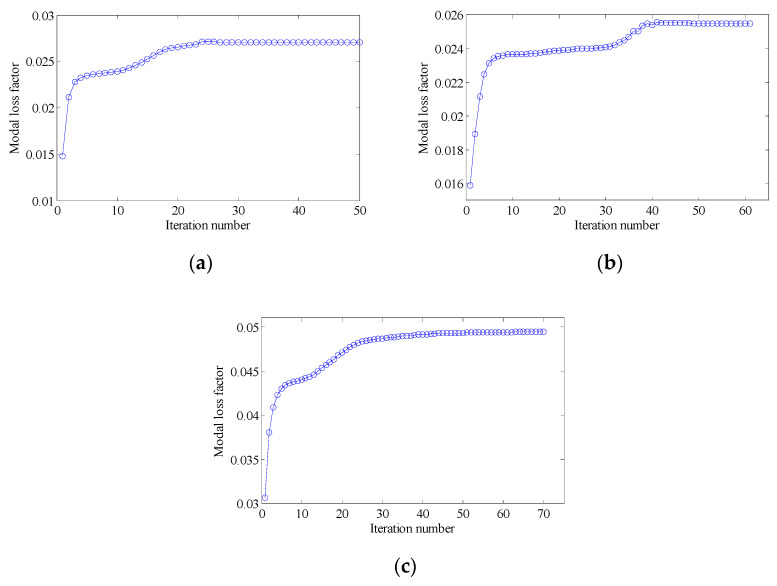
Iteration histories for different optimization objectives: (**a**) Objective 1: maximizing the first modal loss factor; (**b**) Objective 2: maximizing the second modal loss factor; (**c**) Objective 3: maximizing the sum of the first two modal loss factors.

**Figure 22 materials-15-03512-f022:**
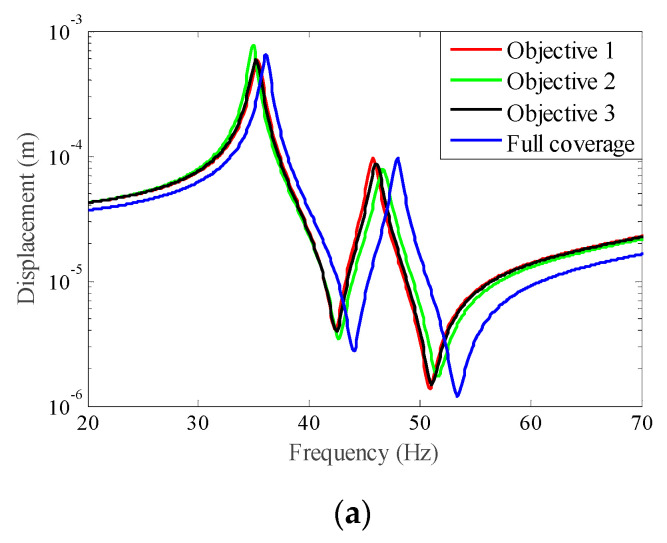
Comparison of the frequency response curves of full coverage structure and optimal designs: (**a**) The frequency response curves from 20 Hz to 70 Hz; (**b**) the detailed description in Mode 1; (**c**) the detailed description in Mode 2.

**Figure 23 materials-15-03512-f023:**
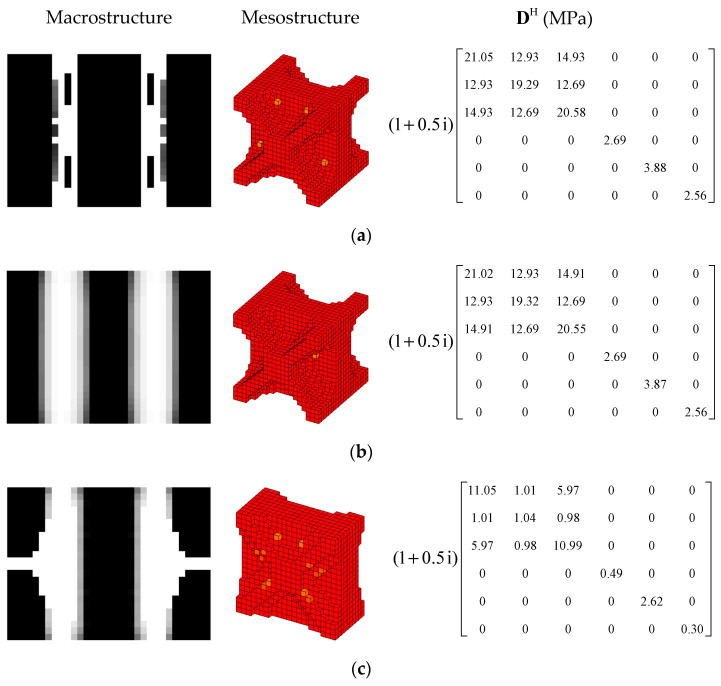
Design results for maximizing the first modal loss factor: (**a**) Case 1: $V_{f}^{\mathrm{MA}}=0.8$ and $V_{f}^{\mathrm{MI}}=0.6$; (**b**) Case 2: $V_{f}^{\mathrm{MA}}=0.6$ and $V_{f}^{\mathrm{MI}}=0.6$; (**c**) Case 3: $V_{f}^{\mathrm{MA}}=0.6$ and $V_{f}^{\mathrm{MI}}=0.4$.

**Figure 24 materials-15-03512-f024:**
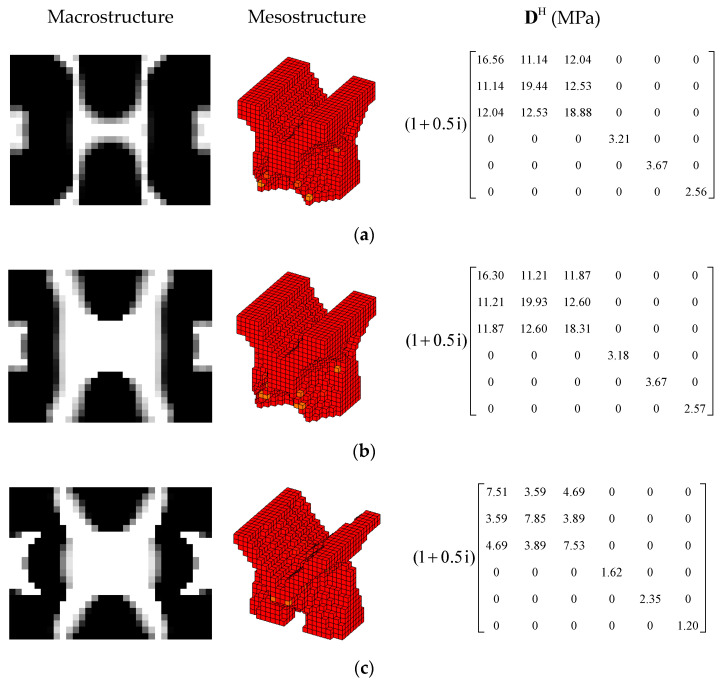
Design results for maximizing the second modal loss factor: (**a**) Case 4: $V_{f}^{\mathrm{MA}}=0.8$ and $V_{f}^{\mathrm{MI}}=0.6$; (**b**) Case 5: $V_{f}^{\mathrm{MA}}=0.6$ and $V_{f}^{\mathrm{MI}}=0.6$; (**c**) Case 6: $V_{f}^{\mathrm{MA}}=0.6$ and $V_{f}^{\mathrm{MI}}=0.4$.

**Figure 25 materials-15-03512-f025:**
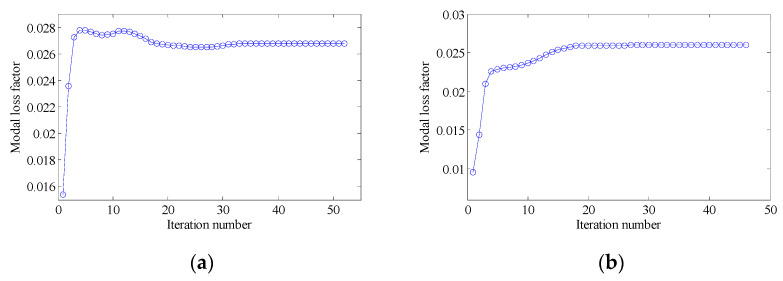
Iteration histories for maximizing the first modal loss factor: (**a**) Case 1: VfMA=0.8 and VfMI=0.6; (**b**) Case 2: VfMA=0.6 and VfMI=0.6; (**c**) Case 3: VfMA=0.6 and VfMI=0.4.

**Figure 26 materials-15-03512-f026:**
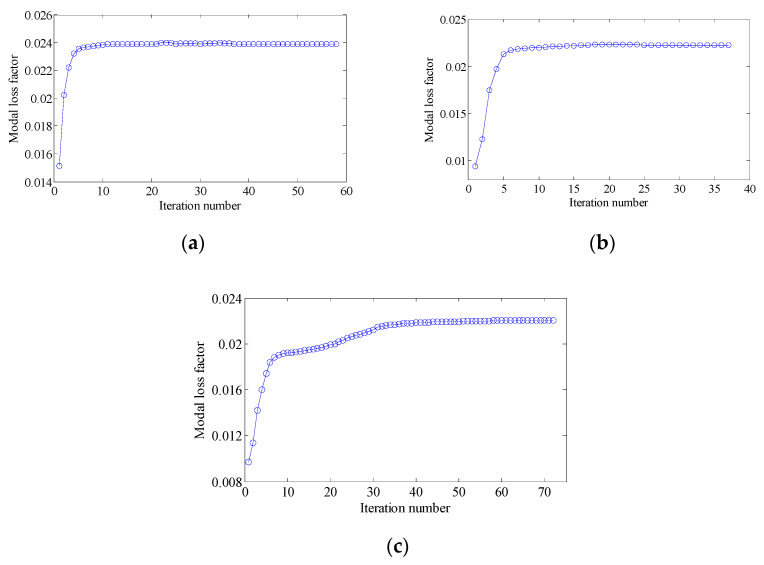
Iteration histories for maximizing the second modal loss factor: (**a**) Case 4: $V_{f}^{\mathrm{MA}}=0.8$ and $V_{f}^{\mathrm{MI}}=0.6$; (**b**) Case 5: $V_{f}^{\mathrm{MA}}=0.6$ and $V_{f}^{\mathrm{MI}}=0.6$; (**c**) Case 6: $V_{f}^{\mathrm{MA}}=0.6$ and $V_{f}^{\mathrm{MI}}=0.4$.

**Table 1 materials-15-03512-t001:** Displacement boundary conditions for the nine load cases.

**1st Load Case**				**2nd Load Case**				**3rd Load Case**			
Nodes	$u_{x}$	$u_{y}$	$u_{z}$	Nodes	$u_{x}$	$u_{y}$	$u_{z}$	Nodes	$u_{x}$	$u_{y}$	$u_{z}$
X = 0	0	Free	Free	X = 0	0	Free	Free	X = 0	0	Free	Free
X = *a*	u	Free	Free	X = *a*	0	Free	Free	X = *a*	0	Free	Free
Y = 0	Free	0	Free	Y = 0	Free	0	Free	Y = 0	Free	0	Free
Y = b	Free	0	Free	Y = b	Free	u	Free	Y = b	Free	0	Free
Z = 0	Free	Free	0	Z = 0	Free	Free	0	Z = 0	Free	Free	0
Z = c	${\bar{\varepsilon }}_{01}X$	0	0	Z = c	0	${\bar{\varepsilon }}_{02}Y$	0	Z = c	0	0	u
**4th load case**				**5th Load Case**				**6th Load Case**			
Nodes	$u_{x}$	$u_{y}$	$u_{z}$	Nodes	$u_{x}$	$u_{y}$	$u_{z}$	Nodes	$u_{x}$	$u_{y}$	$u_{z}$
X = 0	0	Free	Free	X = 0	Free	0	0	X = 0	Free	0	0
X = *a*	0	Free	Free	X = *a*	Free	0	0	X = *a*	Free	0	0
Y = 0	0	Free	0	Y = 0	Free	0	Free	Y = 0	0	Free	0
Y = b	0	Free	0	Y = b	Free	0	Free	Y = b	u	Free	0
Z = 0	0	0	Free	Z = 0	0	0	Free	Z = 0	Free	Free	0
Z = c	0	u	0	Z = c	u	0	0	Z = c	${\bar{\varepsilon }}_{06}Y$	0	0
**7th load case**				**8th load case**				**9th load case**			
Nodes	$u_{x}$	$u_{y}$	$u_{z}$	Nodes	$u_{x}$	$u_{y}$	$u_{z}$	Nodes	$u_{x}$	$u_{y}$	$u_{z}$
X = 0	0	Free	Free	X = 0	0	Free	Free	X = 0	0	Free	Free
X = *a*	u	Free	Free	X = *a*	u	Free	Free	X = *a*	0	Free	Free
Y = 0	Free	0	Free	Y = 0	Free	0	Free	Y = 0	Free	0	Free
Y = b	Free	u	Free	Y = b	Free	0	Free	Y = b	Free	u	Free
Z = 0	Free	Free	0	Z = 0	Free	Free	0	Z = 0	Free	Free	0
Z = c	${\bar{\varepsilon }}_{01}X$	${\bar{\varepsilon }}_{02}Y$	0	Z = c	${\bar{\varepsilon }}_{01}X$	0	u	Z = c	0	${\bar{\varepsilon }}_{02}Y$	u

**Table 2 materials-15-03512-t002:** Material properties of the plate with constrained layer damping treatment.

Layer	Density (kg/m^3^)	Young’s Modulus (MPa)	Poisson’s Ratio	Material Loss Factor
Base plate,	7900	$\mathrm{2.06}\times 10^{5}$	0.3	——
Viscoelastic layer	1200	20	0.495	0.5
Constrained layer	2800	$7\times 10^{4}$	0.3	——

**Table 3 materials-15-03512-t003:** Comparison of the modal loss factors between optimized structure and full coverage structure (bold italics denote the modal loss factor of objective mode).

	Objective 1	Objective 2	Objective 3	Full Coverage Structure
The first modal loss factor	** *0.024* **	0.019	0.023	0.023
The second modal loss factor	0.022	** *0.026* **	0.026	0.024
The sum of the first two modal loss factors	0.046	0.045	** *0.049* **	0.047

**Table 4 materials-15-03512-t004:** The modal loss factors of the optimal designs (bold italics represent the modal loss factor of objective mode).

	The Optimum Design for Maximizing the First Modal Loss Factor			The Optimum Design for Maximizing the Second Modal Loss Factor		
	Case 1	Case 2	Case 3	Case 4	Case 5	Case 6
The first modal loss factor	** *0.025* **	** *0.023* **	** *0.022* **	0.019	0.012	0.013
The second modal loss factor	0.021	0.018	0.019	** *0.026* **	** *0.023* **	** *0.023* **

**Table 5 materials-15-03512-t005:** Comparison of the modal loss factors between optimized structure and full coverage structure (bold italics denote the modal loss factor of objective mode).

	Objective 1	Objective 2	Objective 3	Full Coverage Structure
The first modal loss factor	** *0.027* **	0.021	0.027	0.023
The second modal loss factor	0.021	** *0.025* **	0.023	0.019
The sum of the first two modal loss factors	0.048	0.046	** *0.050* **	0.042

**Table 6 materials-15-03512-t006:** The modal loss factors of the optimal designs (bold italics denote the modal loss factor of objective mode).

	The Optimum Design for Maximizing the First Modal Loss Factor			The Optimum Design for Maximizing the Second Modal Loss Factor		
	Case 1	Case 2	Case 3	Case 4	Case 5	Case 6
The first modal loss factor	** *0.027* **	** *0.026* **	** *0.024* **	0.021	0.019	0.017
The second modal loss factor	0.023	0.019	0.017	** *0.024* **	** *0.022* **	** *0.022* **

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
