# Peer review of "Concurrent Topology Optimization for Maximizing the Modal Loss Factor of Plates with Constrained Layer Damping Treatment"

_materials, 2022, doi:10.3390/ma15103512_

Round 1
Reviewer 1 Report
The paper is written in a very good manner. The topics of the research conducted represent a current engineering issue. The content of the work does not raise any objections and is prepared in a clear manner. The work deserves special recognition. Only a few minor points need to be corrected in the work to make it stronger:
1. in the introduction, the novelty of the present work should be clearly emphasized, in relation to other thematically similar research works.
2. the introduction should include work on optimization: (doi) 10.12913/22998624/61931, 10.1016/j.ifacol.2017.08.2287, 10.1016/j.enbuild.2017.09.065.3. In conclusions there should be a reference to quantitative evaluation of research results and not only qualitative.
Reviewer 2 Report
Paper presents a finite element analysis based approach for optimising the vibration absorption response of a multi-layered medium. key contribution is in the consideration of both meso and macro-scale optimisation parameters for the problem domain. Paper is vague in a lot of areas as highlighted below. These need authors attention for clarification to the reader.
Comments
========
lines 343-352, 400 - 405: What is the difference between the initial guess design 1 and initial guess design 2 beyond the mesh densit? are there any geometric differences in the geometric parameters of the meso-scale visco-elastic layer. This is relevant to the reader for interpreting the results and supporting the claim in lines 400 - 405.
lines 408-420: it appears that for this analysis a different volume fraction is adopted than what was initially used for lines 379 - 384, why is this so? please clarify.
lines 421-424: it is difficult to correlate the data between the figures and respective table. In the figures it is referred to case 1-3 with respective MLF. However, it becomes difficult linking to the tables. Here, it is required to simplify the naming conventions. e.g. case 3 sum of first two MLF does not even correspond to any value on the table (> 0.045)
lines 437-456: here again it is referred to cases, with no clarification of the geometrical parameters of such. Are they same as previously eluded to. Remember, this information was omitted previously. This is confusing and vague to the reader. This needs clarification.
lines 501-522: it is unclear what the final optimised system configuration both at meso and macro scale for the proposed concept.
Reviewer 3 Report
The review is attached.

Round 2
Reviewer 2 Report
Satisfactory updated version of manuscript.
Author Response
Thank you very much for your comments.
Reviewer 3 Report
The manuscript has been
sufficiently improved to warrant publication.
Author Response
Thank you very much for your comments.